# Community-Engaged Research (CEnR) to Address Gaps in Chronic Kidney Disease Education among Underserved Latines—The CARE Study

**DOI:** 10.3390/ijerph20217026

**Published:** 2023-11-06

**Authors:** Janet Diaz-Martinez, Laura Kallus, Harris Michael Levine, Frank Lavernia, Aydevis Jean Pierre, Jessica Mancilla, Ale Barthe, Carlos Duran, Wayne Kotzker, Eric Wagner, Michelle M. Hospital

**Affiliations:** 1Research Center in a Minority Institution, Florida International University (FIU-RCMI), Miami, FL 33199, USA; wagnere@fiu.edu (E.W.); hospitam@fiu.edu (M.M.H.); 2Robert Stempel College of Public Health and School of Social Work, Florida International University, Miami, FL 33199, USA; 3Caridad Center, Boynton Beach, FL 33472, USA; lkallus@caridad.org (L.K.); diabetescontrol@aol.com (F.L.); ajeanpierre@caridad.org (A.J.P.); jmancilla@caridad.org (J.M.); abarthe@caridad.org (A.B.); 4Ker-Twang, Miami, FL 33136, USA; harris@ker-twang.com; 5Florida Kidney Physicians, Boca Raton, FL 33431, USA; cmartinez@flkidney.com (C.D.); wkotzker@flkidney.com (W.K.)

**Keywords:** underserved patients, Latine, CKD awareness and screening, community-engaged research, human-centered design

## Abstract

Ensuring equitable chronic kidney disease (CKD) education for Latine patients with low health literacy and low English proficiency stands as a critical challenge, and the “Caridad Awareness and Education” (CARE) initiative represents our ongoing effort to address this imperative issue. In collaboration with twenty-three patients living with CKD, diabetes and/or hypertension and twelve trained Community Health Workers (CHWs) from diverse Latine subgroups, we conducted a research initiative funded by the National Kidney Foundation. Our primary objective was to co-design and test culturally tailored patient education materials (PEMs) for underserved Latine adults at risk for or diagnosed with CKD. We effectively integrated Community-Engaged Research (CEnR) principles with a Human-Centered Design (HCD) approach to create a range of CKD-PEM prototypes in Spanish. Patient preferences for printed educational materials were clear. They favored printed materials that incorporated visual content with concise text over digital, email, texts, or online resources and personalized phone outreach and the involvement of CHWs. Additionally, patients identified their unwavering commitment to their families as a forceful motivator for caring for their kidney health. Currently, a culturally and linguistically tailored CKD flipchart for one-on-one education, led by CHWs, is undergoing a pilot testing phase involving a sample of one hundred Latine patients at risk for or diagnosed with CKD. This innovative approach signifies a commitment to amplifying the insights and expertise of the Latine community afflicted by kidney health disparities, effectively embracing a CEnR to forge meaningful and impactful CKD-PEMs.

## 1. Introduction

Latines represent the largest ethnic and racial minority group in the U.S. [1]. Approximately 80% live in deprived neighborhoods [2], and roughly 14% of Latine adults have chronic kidney disease (CKD) [3]. They experience worse prognoses and outcomes compared to Whites [4,5,6], including a 1.4 times faster progression to kidney failure [4]. Latines often face systemic stressors that hinder them from receiving proper, evidence-based care that could improve their clinical outcomes [7,8,9,10,11,12,13,14]. Factors such as poverty [7], low health literacy [8,9], acculturation [10], food insecurity [11], and immigration status [12,13,14] have been found to contribute to a higher risk of CKD [15,16]. In comparison to other racial and ethnic groups, Latine individuals from underserved communities with CKD are less likely to seek medical care [17], receive a CKD diagnosis [18,19], and engage in treatment and lifestyle modifications [20,21]. Furthermore, they have less access to innovative and accessible evidence-based kidney health-promoting resources, exacerbating these health inequities [4,9,17,18,19,20,21,22]. Consequently, they have fewer opportunities to understand the complexity of kidney disease, leading to lower adherence to therapies and poorer self-management skills [9,10,13,18,20,21,22]. Additionally, the asymptomatic nature of kidney disease may contribute to patients’ readiness to engage in medical care and education. Many people at risk for CKD have been reported to have low perceived susceptibility to the disease [23,24], and existing patient education resources often fail to adequately convey CKD information, especially for non-English-speaking patients [20,25]. This becomes even more serious for Latine immigrants with lower language acculturation combined with limited health literacy [8,9,10].

The lack of CKD recognition and education leaves Latine patients and families, and the health-care system, unaware of a significant future risk of poor outcomes, including cardiovascular events, the need for urgent dialysis, and a decreased quality of life [26,27,28]. Essentially, education is a necessary part of a comprehensive strategy to improve kidney health outcomes in Latine communities, increasing awareness, enhancing prevention, and promoting the control of known risk factors [20,29,30,31]. Another important yet rarely noted issue is that Latines are a complex and heterogeneous ethnic group comprised of individuals from various cultures, racial/genetic backgrounds, socioeconomic levels, and countries of origin [32]. This diversity also encompasses differences in language, traditions, immigration histories, and health-care access. Understanding these variations is essential for effective health education initiatives [33]. Therefore, ensuring that Latine patients receive culturally and linguistically appropriate educational support is a significant issue in establishing greater social equity in CKD [34,35,36].

One way to address such issues and optimize the delivery of patient-centered CKD care to underserved Latines is to engage the community in kidney research [34]. Thus, in alignment with the National Kidney Foundation’s (NKF) mission to eliminate preventable kidney disease, we conducted “Caridad Awareness and Education” (CARE)—a Community-Engaged Research project aimed at addressing gaps in education among Latine adult communities with or at risk for kidney disease. CARE is an ongoing study funded by the NKF to co-design and test culturally tailored patient education materials (PEMs) for underserved Latine adults with the goal of increasing patient CKD awareness and screening.

## 2. Materials and Methods

Community Description and Location: The study site was Caridad Center (Caridad) located at Boynton Beach, Florida. Caridad, the largest free clinic in Florida, provides comprehensive medical, social, dental, vision, and mental health care to all who are disadvantaged, uninsured, living in Palm Beach County, and unable to obtain these services elsewhere, with approximately 60,000 yearly patient visits. The community primarily served by Caridad consists of immigrant Latine adults (~95%), but it also includes Caribbean Americans (3%), Black/African Americans (1%), and White/Caucasians (1.5%) with a high prevalence of obesity, diabetes, hypertension, and CKD (80% all combined). Nearly all patients live at or below 200% of the U.S. poverty line with household sizes ranging from 5 to 10 members, Additionally, 81% have attained a high school education or less. The majority have limited English proficiency, low health literacy, and a small percentage, though growing (1%), never received formal education in their country of origin. Caridad has a two-decade-long history of training, certifying, and working with Community Health Workers (CHWs) to integrate them into the care of medically underserved and vulnerable communities. CHWs serve as liaisons between the clinic’s medical and social services and their unique communities by identifying community needs, facilitating access to services, and promoting the cultural and linguistic adequacy of the clinic’s services and educational programs.

### 2.1. Study Design and Procedures

Our study integrated Community-Engaged Research (CEnR) [37] with interactive Human-Centered Design (HCD) [38] to support the creation of innovative and person-centered solutions for CKD education and screening. We partnered with CHWs, community leaders affiliated with Caridad, and Latine patients with or at risk for CKD who receive their routine medical care at Caridad. Our approach to participant recruitment harnessed the considerable community outreach and recruitment expertise of the Caridad Center, which has demonstrated remarkable success in effectively engaging with various immigrant Latine communities in the local area and adeptly addressing their specific needs. The study sought the involvement of CHWs and patients who were invited by Caridad staff and had expressed an interest in participating. To be eligible for participation, patients needed to have a confirmed medical diagnosis of diabetes, hypertension, or CKD as recorded in the electronic medical record. Additionally, participants provided informed consent, and, in recognition of their participation, received compensation amounting to $25 for co-design sessions and $30 per hour for focus group involvement.

After receiving a 2.5 h CKD training curriculum, CHWs and patients engaged in focus groups and co-design sessions guided by the research team and an experienced designer group (Ker-twang) with expertise in community educational material design. All research activities including 2.5 h CKD training curriculum, focus groups and co-design sessions were facilitated in Spanish by the research team. Both the research team and the co-designers were proficient in both Spanish and English. Co-designing chronic kidney disease-patient education materials (CKD-PEMs) involved three phases of HCD: inspiration, ideation, and implementation. The inspiration phase aimed to develop a deep understanding of barriers from the perspectives of CHWs, patients, and stakeholders. The ideation phase translated the problems identified in the inspiration phase into potential solutions using multiple brainstorming approaches. CHWs and patients engaged in HCD activities (e.g., drawing, storyboarding, guided conversations) to share unique community needs and held discussions with the research team regarding content, terminology, nutritional differences, and dissemination platforms. The implementation phase focused on systematically and iteratively prototyping and testing CKD-PEM solutions in Spanish. Possible solutions were rapidly prototyped to create tangible representations and uncover potential implementation challenges. CHWs and patients also discussed their preferences for using printed, audiovisual, or digital materials and platforms for dissemination. Additionally, a Community Advisory Committee (CAC), comprising academic experts from FIU/RCMI and community stakeholders, including primary doctors, nephrologists, and health-care practitioners, was convened to review ongoing progress, provide recommendations for CKD-PEM development, and suggest future strategies for increasing CKD education, screening, and treatment. These steps were iterative and progressed with each subsequent design of the materials, testing, and dissemination. The Institutional Review Board of Florida International University approved this study.

### 2.2. Data Collection and Analysis

We collected data on participants’ demographics and patient characteristics related to CKD, such as medical diagnoses, biochemical data, and social determinants of health (including food security, housing, employment, and education level). Qualitative data collection included key themes identified regarding barriers and facilitators of CKD screening and treatment as well as open-ended process evaluation questions regarding CKD-PEMs content relevance and appeal (e.g., terms used, topics, appeal of visuals, etc.) and overall impressions. Qualitative data were analyzed for relevant themes, summarized, and used to guide the iterative processes of co-designing and testing CKD-PEM prototypes and inform materials dissemination [39]. Quantitative data included the overall score of the Patient Education Material Assessment Tool (PEMAT-P) [40] used to evaluate the final CKD-PEM drafts and tracking records of patient CKD screening since the initiation of the study. Recommendations from the CAC were analyzed and implemented throughout the study.

## 3. Results

### 3.1. Participant Characteristics

We engaged 12 Community Health Workers (CHWs), 11 females and one male from various subgroups of the Latine population, representing the patient community at Caridad Center. These subgroups included Cubans, Puerto Ricans, Dominicans, Mexicans, Central Americans, Venezuelans, Colombians, and other South Americans (Table 1). The patients in our study were from Mexico, El Salvador, Honduras, Colombia, Nicaragua, Venezuela, and Chile. Thirteen of them were female and ten were males with a mean age of 55.92 ± 12.44 years. Among the patients, 9 had CKD, 2 had end-stage kidney disease, 17 were diabetic, 14 were hypertensive, and 12 had both conditions. The mean A1c level was 8.35 ± 1.66%, eGFR was 69.51 ± 37.88 mL/min/1.73 m^2^, and the urine albumin-to-creatinine ratio (UACR) was 473.77 ± 294.81 mg/g (Table 1).

### 3.2. Barrier Identification

During the inspiration phase, participants were encouraged to identify barriers to CKD education and screening. These barriers were thematically grouped into different categories, including “difficulties with reading and understanding CKD information”, “uncertainty about who should be tested, how, and how often”, “low technology and numeracy literacy”, “patient fear of asking doctors questions”, “negative emotions upon diagnosis”, “concerns about becoming a burden to their family”, “tendency to self-medicate and use traditional medicine”, “undesirable side effects from medications”, and “lack of trust in the use of online platforms, and concerns about privacy”. Some participants highlighted that undocumented immigrants distrust the health-care system due to fear of deportation. Other identified themes included “poor understanding of medical terminology”, “how to take medications”, and “strong values and beliefs favoring herbal medicine over prescribed medications”.

### 3.3. Solutions Brainstorming

During the ideation phase, participants engaged in brainstorming solutions. The main solutions or facilitators identified were as follows: “materials with gender-neutral images”, “minimal text”, “addition of psychosocial and family support information”, “culturally relatable images and texts”, “conveying a sense of disease progression versus a one-time diagnosis”, “action-oriented content”, “avoidance of overly complex solutions”, and “involvement of CHWs”. Participants also suggested including the phrase, “Hazlo por ellos” (Do it for them), referring to their family, in flyers and posters to help alleviate fears related to CKD testing and become motivated.

### 3.4. Implementation of Solutions

Subsequently, during the implementation phase, participants introduced more solutions. For example, they shared how the social influence of family, friends, CHWs, and other patients can significantly impact their health-care decision making. They perceived CHWs as more trustworthy sources of education compared to internet-based tools or apps, including email, text messages, or WhatsApp. Patients described how CHWs mirrored their community’s demographics and language characteristics, sharing their experiences in dealing with social, structural, and communication challenges. They concluded that printed materials and personalized outreach phone calls from CHWs in Spanish were preferred methods for communicating health education information over text/WhatsApp or email messages. Participants also provided feedback on appropriate font sizes and the use of simpler layperson terms to cater to readers with diverse educational backgrounds. They favored pictures with captions and colorful illustrations to ensure clearer information comprehension, especially for patients with varying literacy levels and illiterate individuals. Furthermore, there was a consensus regarding the amount of information that a material should contain. Participants concluded that tailoring printed materials to specific CKD education goals could enhance their effectiveness in conveying information, engaging the intended audience, and promoting action (behavior change). Thus, they recommended the creation of different CKD-PEMs to address distinct areas of CKD education, aligning with intended goals rather than consolidating all content into a single material. For example, it was recommended to create a general awareness poster (GAP) with the goal of (1) identifying individuals at risk for CKD, (2) explaining how to get tested, and (3) promoting action “to get a CKD test” (Figure 1). Other materials could delve deeper into topics such as risk factors, lifestyle modifications, disease stages, lab result interpretation, treatment goals and CKD patient lab results card indicating the risk of progression (Figure 2 and Figure 3). Table 2 presents a list of the co-created CKD-PEMs and descriptions of the educational content covered in each.

### 3.5. Material Dissemination

Local Latine supermarkets, public transportation stations, barber shops, trailer communities, and food banks were identified as the top locations for disseminating the general awareness posters (GAPs). A CKD flipchart was co-created to be used for CHWs to deliver 1:1 patient education session in the community and clinic visits.

### 3.6. Quantitative Evaluation

The final draft of the newly co-created flipchart for 1:1 education was evaluated using a validated quantitative instrument: the Patient Education Materials Assessment Tool for Printed Material (PEMAT-P). A panel of ten experts fluent in Spanish, including nephrologists, primary care physicians, nurse practitioners, social workers, CHWs, and a registered dietitian from Caridad, assessed the flipchart’s “understandability” and “actionability” using the PEMAT questionnaire. The average score for understandability was 97%, and for actionability, it was 100%.

### 3.7. CKD Screening

Over the first ten months of the study, 147 patients with diabetes and or hypertension underwent CKD screening at the Caridad Center. In all cases (100%), this screening included tests for glomerular filtration rate (GFR) and urinary albumin-to-creatinine ratio (UACR). The utilization of the UACR test for determining the risk of progression exhibited a remarkable increase when compared to the data from the two preceding years, specifically 2018 and 2019. During these two previous years, approximately less than 20% of patients with diabetes and or hypertension had undergone UACR testing.

## 4. Discussion

We collaborated with patients living with CKD or at risk and Community Health Workers (CHWs) from underserved Latine communities in developing different prototypes of CKD-PEMs, adhering to the principles of Community-Engaged Research (CEnR) and Human-Centered Design (HCD). This approach aimed to address gaps in CKD awareness and screening within this community. Patient engagement is a crucial aspect of kidney health promotion initiatives [41,42]. However, it can be challenging to achieve, particularly within underserved immigrant communities characterized by limited English proficiency, low health literacy, and restricted access to health-care services and resources [8,9,10]. Therefore, involving the individuals and communities affected by kidney health disparities is a necessary first step in addressing these barriers [18,20,22]. We actively sought input from marginalized Latine communities and included various ethnic subgroups to achieve our research goals, ensuring inclusivity and relevance. We fostered collaboration among CHWs, patients, health-care providers, community leaders, and researchers as equal partners. This approach facilitated co-learning and capacity building among all stakeholders.

Human-Centered Design (HCD), originally emerging outside of academic settings, has increasingly been incorporated into public health-oriented research [38,43,44,45]. HCD is a creative and iterative problem-solving approach that combines what is desirable from a human perspective with what is feasible. It provides a structured process for achieving shared goals with steps to translate insights into action (inspiration, ideation, and implementation). By involving users in the design process, HCD can lead to interventions more likely to be sustained over the long term. For instance, Kia-Keating et al. [44] integrated HCD with community-based participatory research to address violence-related inequalities among young Latines, while Pillsbury et al. [45] employed HCD to develop and evaluate implementation strategies for strengthening hypertension referral networks in western Kenya. HCD’s rapid prototyping aligns with our study’s Community-Engaged Research approach and principles. Our methodology enabled CHWs and patients to advocate for themselves and their community, valuing their personal experiences. Participants developed a sense of ownership and investment in the solutions they co-created. One of the most significant lessons learned from this experience was that CHWs and patients, who often felt powerless to change health-related barriers and problems in their community, realized they could take action to address issues affecting kidney health.

In our study, participants showed a strong preference for printed materials and phone calls over audio or videos. Printed materials were considered more convenient for long-term reference, allowed for multiple readings, were less distracting than videos, and were more accessible to those lacking technology access. Phone calls were perceived as more personal, which is supported by previous research in the USA [46,47]. Participants also emphasized the use of minimal text and more visual aids, which were also previously reported [48]. Additionally, patients preferred to receive CKD education from CHWs, highlighting their role as trusted community educators and agents of change. This finding aligns with existing research [49,50,51] and our previous experience in this community [52], which underscored the effectiveness of CHWs in addressing the unique challenges faced by individuals experiencing poverty and social disparities.

Participants’ preference for involving their families in CKD education, as expressed by the term “Do it for them”, has been reported by other authors. Novick et al. [53] found that patients living with dialysis were motivated to continue treatment for the sake of their families. This underscores the importance of family involvement in health-care decisions among Latines. Surprisingly, in the design of the CKD patient lab results card, participants decided to use an adaptation of the original “CKD heat map” [54] and preferred a graphic representation where risk increases from left to right but decreases from bottom to top. Although this is the opposite of the original version, other authors have recommended designing health communication heat maps with the most desirable outcome at the bottom and the least desirable at the top [55]. It is noteworthy that even though this study aimed to increase CKD awareness and screening at the patient level, it also facilitated the recognition of CKD and the utilization of UACR testing. Underutilization of albuminuria testing has been recognized as a barrier to identifying CKD at the primary care level, representing a missed opportunity to improve kidney health equity among insured US adults [56].

At the conclusion of the study (Year 1), the Community Advisory Committee (CAC) drafted a strategic plan and recommendations for increasing CKD testing, education, and proper treatment among high-risk patients. These recommendations included initiating a CKD surveillance system, implementing quality assurance measures for CKD recognition, conducting systematic patient educational workshops co-facilitated by CHWs and kidney health-care professionals, establishing a multidisciplinary CKD clinic staffed with primary doctors, nephrologists, dietitians, social workers, case managers, kidney health navigators, and CHWs, and integrating the use of Motivational Interviewing (MI) principles [57] to address patient resistance and promote positive behavioral changes. Additionally, a pilot community intervention testing the acceptability and preliminary efficacy of one-on-one CKD education led by CHWs in a sample of 100 patients with or at risk for CKD is currently underway (Year 2). This initiative, funded by the National Kidney Foundation for its second year, includes 20 h of curriculum training for CHWs on using the co-designed flipchart as a tool for CKD education and applying MI principles to enhance patient engagement and promote effective CKD self-management. Data collection for this pilot intervention includes CHW satisfaction, confidence in delivering one-on-one CKD education, knowledge of MI principles, patient pre/post knowledge, education acceptability, and preliminary data on efficacy, such as the number of patients referred to the CKD clinic and their attendance.

Our study exhibits several notable strengths. Firstly, our methodology facilitated the co-creation of CKD-PEMs that were specifically tailored to address the unique needs of underserved patients with low health literacy and limited proficiency in English. Secondly, our study achieved the inclusion of Latines from diverse subgroup populations, which are often underrepresented in research. Thirdly, the evaluation of the CKD flipchart for one-on-one education demonstrated its comprehensibility and actionability, and it is currently undergoing testing in a pilot intervention to assess its efficacy. Nevertheless, certain limitations warrant consideration. All sessions were conducted in Spanish, and the materials were exclusively developed in Spanish, potentially constraining the generalizability of our findings to non-Spanish-speaking communities.

Our study holds significant implications for both research and practice. In the realm of research, we underscore the value of integrating CEnR with interactive, Human-Centered Design to foster the development of innovative, person-centered solutions for CKD education and screening in underserved communities. In practical terms, our findings emphasize that patients exhibit a preference for printed materials featuring visual content and concise text, along with personalized phone outreach and the involvement of CHWs, as opposed to digital, email, or online resources. Patients cited their strong commitment to their families as a motivating factor for maintaining their kidney health. These insights can be taken into consideration by organizations working with patients of low literacy and English proficiency, aiming to enhance the acceptability and effectiveness of their health educational materials.

## 5. Conclusions

Our innovative approach prioritizes the lived experiences and expertise of our underserved Latine community in co-creating printed CKD-PEMs. The integration of HCD with CEnR in developing CKD-PEMs was essential for the community to identify critical issues and explore actionable solutions. This approach enhances the likelihood of creating meaningful, relevant, and impactful educational solutions that resonate with the affected community, ultimately leading to positive behavioral change and improved health outcomes. Further research will build upon these findings and explore the impact of culturally tailored CKD education on early detection and CKD prevention in Latine populations. It is also crucial to continue considering the role of sub-ethnicities within the Latine community rather than treating the group as homogeneous. Future studies will continue to investigate sub-ethnic similarities and differences to determine the extent of adaptation needed for culturally tailored materials.

## Figures and Tables

**Figure 1 ijerph-20-07026-f001:**
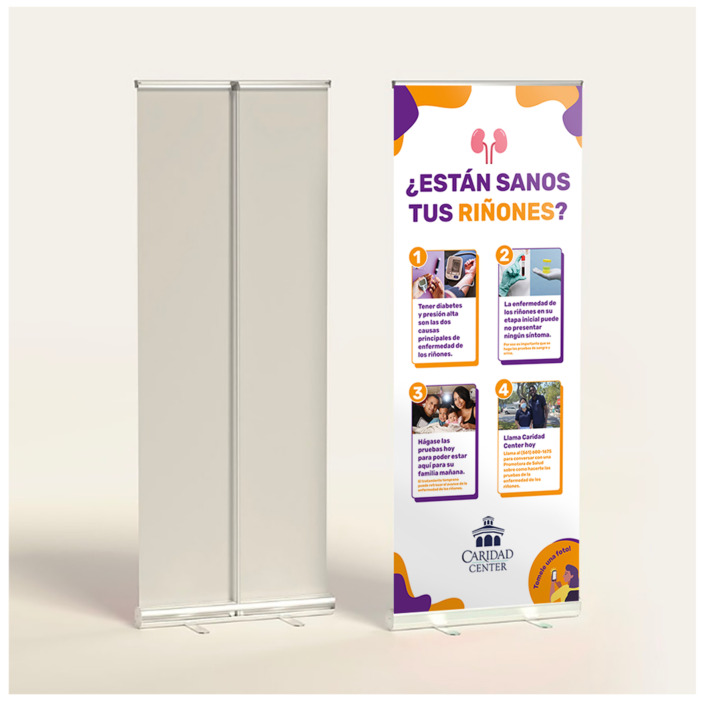
General awareness poster in Spanish promoting CKD screening among Latines with diabetes and hypertension.

**Figure 2 ijerph-20-07026-f002:**
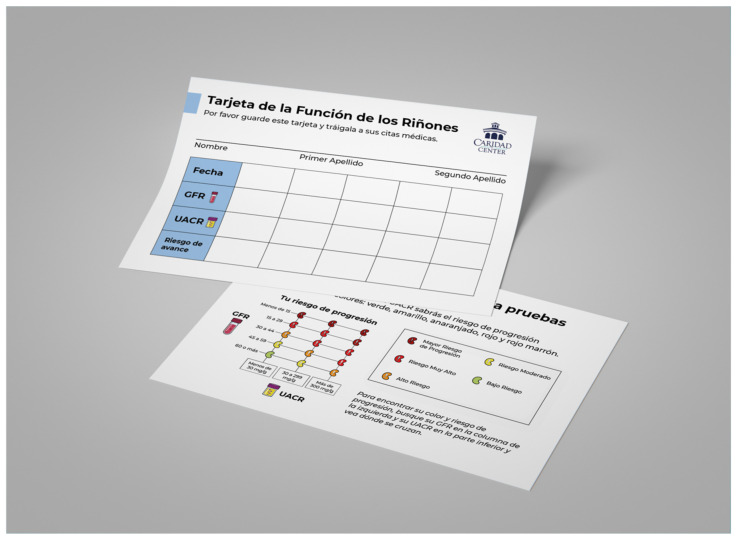
CKD patient card in Spanish, explaining the risk of progression based on patient’s laboratory results for estimated glomerular filtration rate (GFR) and urine albumin-to- creatinine ratio (UACR).

**Figure 3 ijerph-20-07026-f003:**
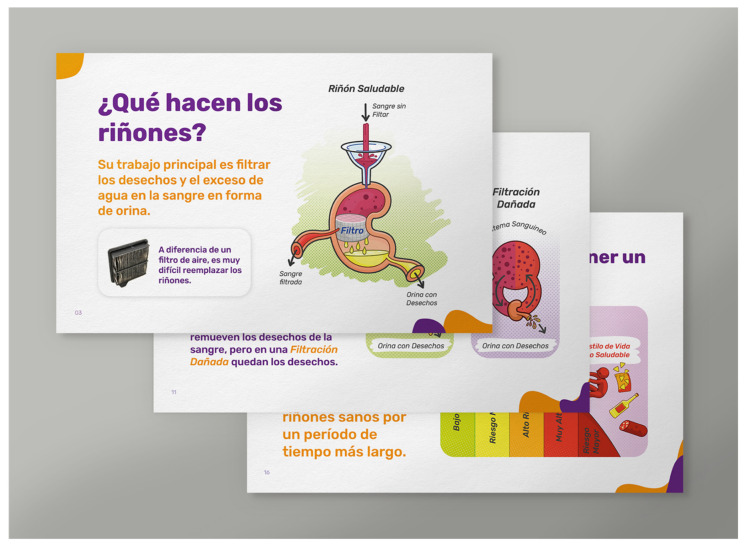
Imagens co-designed by patients and CHWs explaining the function of the kidneys and how they can be damaged by uncontrolled diabetes, hypertension, and unhealthy lifestyles.

**Table 1 ijerph-20-07026-t001:** Characteristics of participants.

CHW (*n* = 12)	
Variable	
Age (Years)	44.6 ± 12.5
Gender	Female 91% (11)
	Male 8.3% (1)
Country of Origin	Mexico 25% (3)
	Guatemala 16.6% (2)
	Cuba 16.6% (2)
	Puerto Rico 16.6% (2)
	Venezuela 8.3% (1)
	Argentina 8.3% (1)
	Chile 8.3% (1)
Patients (*n* = 23)	
Variable	
Age (Years)	55.92 ± 12.44
Gender	Female 56.5% (13)
	Male 43.4% (10)
Country of Origin	Mexico 47.8% (11)
	Salvador 13% (3)
	Honduras 8.7% (2)
	Colombia 13% (3)
	Nicaragua 4.3% (1)
	Venezuela 8.7% (2)
	Chile 4.3% (1)
Educational Level (Less than High School Diploma)	100%
Illiterate	16.6%
Food Insecurity	100%
Level of Poverty (Under 200% Federal Level of Poverty)	100%
Diabetes	74%
Hypertension	60%
CKD/ESRD	48%
A1C % ^1^	8.35 ± 1.66
eGFR (mL/min/1.73 m^2^) ^2^	69.51± 37.88
UACR ^3^	473.77 ± 294.81

^1^ A1C = glycosylated hemoglobin ^2^ estimated glomerular filtration rate ^3^ urine albumin-to-creatine ration.

**Table 2 ijerph-20-07026-t002:** Chronic kidney disease (CKD) education content covered by patient education materials (PEM) prototypes.

CKD-PEM Prototypes	Content area Covered					
	Basics of CKD	Risk for Development and Progression	How to Get Tested	Lab Results Interpretation	Treatment Goals	Lifestyle Modification
General Awareness Posters			x			
Outbound Calls Scripts for CHWs		x	x			
Patient Brochures	x	x	x	x	x	
CKD Patient Lab Results Card ^1^				x	x	
FlipChart for 1:1 Education	x	x	x	x	x	x

^1^ CKD: chronic kidney disease, 1:1 (patient: community health worker indicating one-on-one education modality). The letter “X” within the table indicates content covered on the CKD-PEM prototypes.

## Data Availability

The data presented in this study are available on request from the corresponding author. The data are not publicly available as we only asked informed consent from the participants to share data with other researchers on request.

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
