# Peer review of "Community-Engaged Research (CEnR) to Address Gaps in Chronic Kidney Disease Education among Underserved Latines—The CARE Study"

_ijerph, 2023, doi:10.3390/ijerph20217026_

Round 1

Reviewer 1 Report

Comments and Suggestions for Authors

Were all CHWs and patients included female? Was that by design? If not, why aren’t the male characteristics mentioned?

Was the entire study conducted in English, esp. considering the LEP community participants? Were there some meetings and/or facilitations done in English & others in Spanish (e.g., 2.5 hr CKD training curriculum for CHWs, assuming some are from the community and have LEP also; focus groups w/patients)? Were all printed materials in English & Spanish or Spanish only? Were phone calls that participants preferred done in English & Spanish? The above was unclear to me so I suggest adding more explicit detail about this.

If citations should appear in numerical order, reference 41 is out of order (listed on page 8) so might want to renumber some of the preceding citations to account for that. If citations don't need to be in numerical order as per the journal, then it is okay.

Author Response

Manuscript ID: ijerph-2673455

Title: Community-Engaged Research (CEnR) to Address Gaps in Chronic Kidney Disease Education Among Underserved Latines.- The CARE study.

Response to reviewer 1:

We appreciate the thorough review and suggestions provided by the reviewer 1, and we have responded to each of their concerns and followed their recommendations. Please find our responses below. Changes in the manuscript were highlighted in yellow for easy reference.

REVIEWER 1

Comment 1: Were all CHWs and patients included female? Was that by design? If not, why aren’t the male characteristics mentioned?

Answer 1: No, the study included 11 CHWs female and 1 male. Patients were 13 female and 10 males. This was corrected in lines #159 and #164 and detailed better table (line # 247)

Comment 2: Was the entire study conducted in English, esp. considering the LEP community participants? Were there some meetings and/or facilitations done in English & others in Spanish (e.g., 2.5 hr CKD training curriculum for CHWs, assuming some are from the community and have LEP also; focus groups w/patients)?

Answer 2: All study activities including CKD training curriculum, focus groups and co-design sessions were facilitated in Spanish. Both the research team and the co-designers were proficient in both Spanish and English (added in lines #118-121)

Comment 3: Were all printed materials in English & Spanish or Spanish only? Were phone calls that participants preferred done in English & Spanish? The above was unclear to me so I suggest adding more explicit detail about this.

Answer 3: All printed materials and phone calls were in Spanish. This has been specified in lines # 130 and 201.

Comment 4: If citations should appear in numerical order, reference 41 is out of order (listed on page 8) so might want to renumber some of the preceding citations to account for that. If citations don't need to be in numerical order as per the journal, then it is okay.

Answer 4: We corrected this error in the reference numbers to appear in numerical order. Previous reference # 41: “Miller WR, Rollnick S. Ten things that Motivational Interviewing is not. Behav. Cognit. Psychother. 2009, 37, 129-140 was corrected to indicate current reference #57. And, Current reference #41. “Lunardi LE, Hill K, Xu Q, Le Leu R, Bennett PN. The effectiveness of patient activation interventions in adults with chronic kidney disease: A systematic review and meta-analysis. Worldviews Evid Based Nurs. 2023 Jun;20(3):238-258”

Reviewer 2 Report

Comments and Suggestions for Authors

Thank you for the opportunity to review this manuscript. The authors designed a programme focusing on CKD education and screening for the Latine community. Overall, the approach and results are inspiring, and the manuscript is well written.
Several comments

Page 3 Line 132 - How are these patients identified and recruited? Is there any assessment for their commitment to the project prior to recruitment? Are they paid or rewarded in any way?
Page 4 Table 1 - Add ‘years’ after age in CHW
Page 4 Table 1 - The percentage for food insecurity is missing
Page 5 Line 206, Is there any previous number for comparison to evaluate the increase of people who underwent CKD screening.

Thanks.

Author Response

Manuscript ID: ijerph-2673455

Title: Community-Engaged Research (CEnR) to Address Gaps in Chronic Kidney Disease Education Among Underserved Latines.- The CARE study.

Response to reviewer 2:

We appreciate the thorough review and suggestions provided by the reviewer 2, and we have responded to each of their concerns and followed their recommendations. Please find our responses below. Changes in the manuscript were highlighted in yellow for easy reference.

REVIEWER 2

“Thank you for the opportunity to review this manuscript. The authors designed a programme focusing on CKD education and screening for the Latine community. Overall, the approach and results are inspiring, and the manuscript is well written”.

Several comments

Comment 1: Page 3 Line 132 - How are these patients identified and recruited? Is there any assessment for their commitment to the project prior to recruitment? Are they paid or rewarded in any way.

Answer 1: Our approach to participant recruitment harnessed the considerable community outreach and recruitment expertise of Caridad Center, which has demonstrated remarkable success in effectively engaging with various immigrant Latine communities in the local area and adeptly addressing their specific needs. The study sought the involvement of CHWs and patients who were invited by Caridad staff and had expressed an interest in participating. To be eligible for participation, patients needed to have a confirmed medical diagnosis of diabetes, hypertension, or CKD as recorded in the electronic medical record. Additionally, participants provided informed consent and, in recognition of their participation, received compensation amounting to $25 for co-design sessions and $30 per hour for focus group involvement. This information has been added (page 3, lines #104-#114)

Comment 2: Page 4 Table 1 - Add ‘years’ after age in CHW.

Answer 2: “Years” has been added to Table 1 after age.

Comment 3: Page 4 Table 1 - The percentage for food insecurity is missing.

Answer 3: Food insecurity (100%) has been added to Table 1

Comment 4: Page 5 Line 206, Is there any previous number for comparison to evaluate the increase of people who underwent CKD screening?

Answer 4: In line # 241 the following information was added “….. Over the first ten months of the study, 147 patients with diabetes and or hypertension underwent CKD screening at Caridad Center. In all cases (100%), this screening included tests for Glomerular Filtration Rate (GFR) and Urinary Albumin-to-Creatinine Ratio (UACR). The utilization of the UACR test for determining the risk of progression exhibited a remarkable increase when compared to the data from the two preceding years, specifically 2018 and 2019. During these two previous years, approximately less than 20% of patients with diabetes and or hypertension had undergone UACR testing.”